# Phase Inversion Gelation Process and Additive Effects on Hydrogel Film Properties of Cotton Cellulose

**DOI:** 10.3390/gels10010034

**Published:** 2023-12-31

**Authors:** Ayano Ibaraki, Takaomi Kobayashi

**Affiliations:** Department of Science of Technology Innovation, Nagaoka University of Technology, Niigata 940-2188, Japan; a_ibaraki@stn.nagaokaut.ac.jp

**Keywords:** cotton cellulose hydrogel, phase inversion, cellulose fibers, water retention

## Abstract

During the preparation of cotton cellulose hydrogels using the phase inversion gelation method of *N*,*N*-dimethylacetamide/LiCl solution under ethanol vapor, acetone (AC), methyl ethyl ketone (MEK), or diethyl ketone (DEK) were added as additives, and their gelation state and the properties of the resulting hydrogels were evaluated. Adding the ketones to the cellulose solution caused an increase in the gelation time, but the solution viscosity decreased, indicating that the cellulose tended to aggregate in the solution. Among the hydrogels prepared by adding ketones, the water content was as high as 2050%, especially for AC and MEK. In these hydrogels, cellulose formed an agglomerated fibrous network of a few micron widths, forming a tuft-like entrapment space of about 10 to 100 μm size. The structure surrounded water and held it in the hydrogels. The FTIR results showed that the water, which formed hydrogen bonds, was retained within the hydrogel network. This structural configuration was determined to be conducive to maintaining the gel state against external deformation forces, especially in the case of the addition of MEK.

## 1. Introduction

Growing environmental concerns regarding sustainability have led to an increased focus on bio-based and environmentally friendly materials [1,2,3]. In response to social calls for sustainable resource utilization and the requirement to mitigate carbon emissions, biomass-based natural polymers are increasingly replacing synthetic polymers derived from petroleum-based materials [4]. Cellulose, as a naturally occurring resource, stands out as the most abundant natural polymer on Earth, characterized by its sustainable availability and renewability, along with distinctive properties like biodegradability. Within the cellulose source, lignocellulosic biomass is a prominent natural and carbon-neutral resource. Particularly, wooden biomass is in high demand, especially in the construction and paper industries. However, there is concern about the depletion of wood resources due to deforestation. In addition, from the perspective of the sustainable conservation of forest resources, it has become difficult to secure sufficient wood resources in recent years. To address the growing global demand and consumption of cellulose, there is a need to explore the use of non-wooden biomass. Cultivating such non-wooden biomass, which is managed and characterized by short harvesting periods, ensures a stable resource supply. Also, wooden pulp, constituting 40–50% of the cellulose in plant cell walls, is converted to cellulose mainly through a refining process using chemical treatments. In contrast, among non-woody biomass, cotton stands out as a promising market candidate for cellulose resources with an impressive 90% cellulose content, requiring no extraction process and making it a further environmentally friendly material [5,6,7]. The stable productivity of such characteristics in cotton cellulose results in great features for a sustainable resource. Despite these advantages, cotton is currently not widely utilized as a functional material outside of textile industry applications. This limitation arises from the challenge that, unlike other petroleum-derived polymers, cellulose is insoluble in water and most organic solvents [8,9]. Additionally, it does not exhibit flow when heated [10,11], requiring specialized solvent compositions for processing [8,9,10].

In response to these drawbacks, various solvents capable of dissolving cellulose have been investigated [12,13,14]. For example, *N*,*N*-dimethylacetamide (DMAc) and LiCl are well-known solvents that dissolve cellulose. However, due to their high boiling points and the technical problems in their efficient removal to make cellulose materials, there has not been much progress in the functionalization of cellulose. In contrast to this, a novel approach to prepare cellulose hydrogel films from cellulose-DMAc/LiCl solutions using phase inversion gelation has been discovered [15]. In this method, the phase inversion gelation method results in the sol–gel transformation of cellulose solution. During the process, the cellulose solution makes contact with water vapor or alcohol vapor, reducing the cellulose solubility. This reduction in solubility leads to gelation, as the water vapor concentration in the DMAc/LiCl solution increases [16]. The cellulose hydrogel films fabricated from the DMAc/LiCl solution have outstanding water absorbance ability with high mechanical strength. The process involves the solidification of cellulose by introducing a cellulose solution in a DMAc/LiCl solvent into contact with a solvent such as water or alcohol. This process constructs a hydrogel 3D network without the need for chemical modification or crosslinking agents for the gelation of cellulose. As a result, the application of such hydrogel films was demonstrated to have excellent biocompatibility [14] and affinity for skin cells [17] and was used as an instrument for drug release in several medicines [18,19], cosmetic applications [20], and tissue engineering [21].

To utilize cellulose hydrogel films for biomedical applications, controlled morphology plays an important factor in enhancing their properties [22,23,24]. These hydrogels allow structural modifications through the chemical modification of polymer and the choice of crosslinking methods, influencing property changes in water retention ability and mechanical strength, enhancing controlled drug releasing and cell affinity [25,26]. For instance, decreasing the concentration of LiCl in the DMAc/LiCl solution changed the aggregation structure of the dissolved cellulose, and the cellulose in the resulting hydrogel showed a tendency to arrange in a fibrous manner, improving cell affinity [15]. Also, cellulose concentrations in the solution have an effect on the water swelling and mechanical properties of hydrogels [27]. Thus, the aggregation behavior or concentration of polymer in cellulose solution are essential parameters for controlling the physicochemical properties of the resultant hydrogel. On the other hand, ketones are known as anti-solvents for dissolved cellulose, and the addition of ketones to various solvent systems such as ionic liquid (ILs) and tetrabutylphosphonium hydroxide (TBPH) has been reported to alter the aggregation behavior of cellulose in solution [28,29,30]. However, the impact of ketone addition to the DMAc/LiCl solvent system has not been reported, and its effects on the structure and properties of the hydrogels are still unknown. Understanding the effects of ketone addition to the DMAc/LiCl solvent on the gelation process may be crucial for improving the controllability of cellulose hydrogel properties for biomedical applications.

Against this background, we aimed to examine the effect of additives on the hydrophilic properties of cellulose hydrogel films produced when small amounts of anti-solvents are added within a DMAc/LiCl solution of cotton cellulose. Initially, cotton fibers, serving as the raw material for cellulose, were dissolved in a DMAc/LiCl direct solvent; then, three kinds of ketones were introduced into the solution subjected to solution–gelation transition through the phase inversion gelation process. Specifically, the gelation behavior and hydrogel properties of cellulose hydrogels in the presence of additives that are incompatible with cellulose in DMAc/LiCl solutions of cotton cellulose are discussed in this paper.

## 2. Results and Discussion

### 2.1. Additive Effects on Cellulose Gelation

Following solvent exchange treatment on cotton fiber with water, ethanol, and DMAc, respectively, a cotton cellulose solution with a cellulose: DMAc: LiCl weight ratio of 1:93:6 was prepared by mixing and stirring for 1 month, resulting in a clear and colorless solution. Subsequently, acetone (AC), methyl ethyl ketone (MEK), and diethyl ketone (DEK) were introduced as additives at 0.1 to 0.5 moles per gram of cellulose. As shown in Figure 1, adding these ketones resulted in cellulose precipitation, forming a clear gel with AC and a white precipitate with MEK and DEK. Indeed, acetone is widely recognized as an antisolvent for dissolved cellulose in certain solvents [28,29,30]. In the case of ILs, hydrogen bonds between cellulose and ILs can be disrupted through the exchange of cellulose-bonded anions induced by the strong attraction of polar solvents. Similarly, in the case of the DMAc/LiCl solvent used in this study, it results in the cleavages of bonds between cellulose fibers and solvents, inducing fiber aggregation. However, the aggregated cellulose fibers were partially re-dissolved to the DMAc/LiCl solution because DMAc can solvate with the ketones. Actually, when the additive amounts were up to 0.3 mol of AC and MEK, and also up to 0.2 mol of DEK, the cellulose-DMAc/LiCl solutions demonstrated homogeneity with no sedimentation, as observed in the inserted pictures in Figure 1. However, in clear solutions containing ketones, the fiber would be dissolved in DMAc/LiCl in an aggregated state. On the other hand, concentrations beyond these limits resulted in non-homogeneous solutions, with agglomerated fibers remaining in the solution.

Figure 1 shows the viscosity results of the clear cellulose solutions at various additive amounts. As the shear rate of the viscometer varied from 0.1 to 1000 1/s, the solution viscosity displayed a tendency to decrease, exhibiting thixotropic behavior. In the absence of additives, the viscosity was notably high at 3.8 Pa·s at a shear rate of 0.1. When the ketones were added, the viscosity values at a shear rate of 0.1 1/s tended to decrease as seen in (B)–(C). Similarly, due to the dilution effect resulting from the addition of additives, we also studied the viscosity change in DMAc with the additional DMAc solvent, as depicted in (a). Clearly, the addition of DMAc and AC in (A) and (B), respectively, showed a similar viscosity trend, while the addition of MEK and DEK in (C) and (D) both exhibited lower viscosity values. In this case, it can be inferred that the presence of MEK and DEK in DMAc/LiCl dissolved the celluloses in a shrinking state of the molecular chains. Especially, the viscosity values dropped to 0.2 Pa·s in the cases of MEK 0.3 mol and DEK 0.2 mol solutions. However, when 0.4 moles of AC and MEK and 0.3 moles of DEK were added, the white precipitation of cellulose occurred, as shown in the photo in Figure 1. At each dynamic shear rate (s^−1^), the solutions with Non additive (NA) and AC exhibited a rapid decrease in viscosity upon the onset of shear. Conversely, in the cases of MEK and DEK, a notable decrease in viscosity was not observed until shear rate leached to approximately 10 s^−1^. These findings indicate that MEK and DEK require higher energy to flow cellulose fibers, suggesting different cohesion properties among different ketones. In a comparison of the three additives, the shear viscosity tended to decrease with an increase in alkyl chain length. This trend indicates that the cohesiveness of cellulose in solution is in the order DEK > MEK > AC > NA.

When such solutions are exposed to ethanol vapor, cellulose gradually becomes less soluble, and gelation occurs [15,16]. Gelation behavior in cellulose hydrogel was observed under an ethanol atmosphere while measuring dynamic storage modulus G′ and loss modulus G″. Figure 2 shows the time variation in each elastic modulus. The phase transformation process of cotton cellulose from the DMAc/LiCl solution to the gel state was observed for time variation using a dynamic viscoelastic apparatus, since a gel-like solid-containing solvent is formed in the solution. So, the transformation process can be observed in the increase in the value of viscoelasticity, as shown in Figure 2. The value of the storage modulus (G′) was about a few Pa in the cotton cellulose solution. However, when gelation occurs, it increases several hundredfold with time as it repels mechanical deformation forces. The 1% cellulose solution without additives exhibited a G′ value of approximately 4 Pa even under a 0.1 strain % deformation. However, this value gradually increased after 10 min of standing, reaching almost 1000 Pa at 80 min under ethanol vapor. The G″ increased slightly from 20 Pa to 50 Pa during the same period. The onset of polymer crosslinking triggered this rapid increase in the G′, and the time the increase began is referred to as the crosslinking time. The addition of ketones extended the time required for cellulose to undergo crosslinking in the solution. Specifically, adding 0.3 mol of AC took 4.5 times longer time than NA. The crosslinking time for all three additives tended to be delayed as the concentration of the additive increased, likely attributable to a decrease in the cohesive effect of the fibers due to dilution. On the other hand, when comparing the three ketones, AC, with the shortest alkyl chain, took the longest time to crosslink, followed by MEK, DEK, and NA. This result indicated that the presence of the longer alkyl chain of the ketone in the solution resulted in faster solvent exchange.

Figure 3 shows the time variation of tan *δ*, the value of the ratio G′/G″ for Figure 2. As crosslinking progresses, the fiber gradually loses fluidity and transform into an elastic gel. This sol–gel transition is characterized by the G′ and the loss modulus (G″) crossover, defining the gelation time. Clearly, the value of tan *δ* tends to decrease toward 1 with increasing exposure time. As listed in Table 1, the gelation time without additives was determined to be 16.1 min. With an increase in the amount of AC to 0.1 mol, 0.2 mol, and 0.3 mol, the time extended to 24.3, 36.4, and 51.0 min, respectively. This indicates that, in the case of AC addition, the gelation time is delayed with an increase in the amount of AC. In contrast, in the case of MEK, gelation tended to be faster than in the AC system, and even faster in DEK, but not faster than in the case with NA. This trend was the same as the crosslinking time, suggesting that adding a longer alkyl chain of ketone not only promoted fiber aggregation but also led to a faster solvent exchange between DMAc/LiCl and ethanol.

### 2.2. Properties of Cotton Cellulose Hydrogels

The cotton cellulose hydrogels produced in an alcohol atmosphere displayed transparency in films with a thickness of 0.8 mm, as shown in the appearance pictures in Figure 4. Both hydrogels have the ability to retain water, and as shown in Figure 5, the cotton hydrogel showed a water content of 1900% on a dry basis.

The hydrogels with AC and MEK exhibited an increased water content with higher additive amounts, especially for AC 0.3, and MEK 0.3 exhibited about a 2050% water content. In contrast, the hydrogel formed with the addition of DEK had less water retention compared with the cotton hydrogel. The viscosity measurements (Figure 1) revealed a significant reduction in viscosity and gelation kinetics (Figure 3), suggesting that the cellulose molecules are more aggregated in DMAc/LiCl in the presence of DEK. Thus, the gelatinization of the cellulose tended to occur more quickly when DEK was added. The results suggest that the accelerated crosslinking of the cellulose led to a decrease in water retention in the hydrogel.

The SEM analysis was conducted to observe the fiber network structure of each hydrogel, as depicted in Figure 4. The SEM images of surface morphology were measured in wet conditions with freezing at −20 °C under the vacuum by using a Peltier-driven SEM cooling stage. On the other hand, the freeze-dried hydrogel was used for the cross-section morphology. The structure of the hydrogel on the surface of the cellulose hydrogel without additives appeared extremely heterogeneous, consisting of cellulose aggregated to about 1 μm or less. This cellulose network formed a shelf-like depression with a width ranging from several 10 to 50 μm surrounding the water. In the cross section of the NA hydrogel, the depressions formed by the aggregated cellulose network were 70–100 μm in size larger than those observed on the surface. This suggests significant cellulose aggregation at the hydrogel surface.

In the case of the AC hydrogels, the assembly of cellulose fibers seemed to surround a larger body of water, ranging from 10 to 30 μm in size for AC 0.1. As the amount of AC increased, the aggregation of cellulose fibers seemed to intensify, leading to an improvement in homogeneity in the surface structure. This was due to the thicker fiber assembly surrounding larger water clumps. When the AC was added with 0.2 mol to the cellulose unit weight, the surface morphology displayed more regularly sized cellulose aggregated fibers, with about 20–40 μm in the size of that depression. The cellulose network in the cross-sectional view appeared to be a dense cellulose wall surrounding the water, with an overlapping structure of 20–40 µm in size. This structure resembled an entrapped bag of cellulose fibers stacked in tufts. At 0.3 mol of AC, the surface depressions became more distinct and somewhat wider compared with 0.1 mol and 0.2 mol. The shelf-like structure, forming a water lump surrounding cellulosic fibers, presents a suitable structure to retain a greater capacity of water. This typical homogeneous porous structure favors the release properties and water retaining ability of the cellulose-based hydrogels and supports the possibility of controlling the morphology of the porous networks to obtain versatile cellulose-based materials [22,31].

MEK and DEK hydrogels showed a porous structure consisting of thicker fibers compared with NA and AC. The MEK 0.2 hydrogel tended to have slightly larger circular surface depressions with 10–30 μm, and the width of the agglomerated cellulose fibers was thicker than that observed in AC 0.2. This result was easily inferred from the results of cellulose agglomeration promoted by the effect of MEK addition. In the SEM analysis of the surface morphology, the impacts of MEK addition at concentrations of 0.1–0.3 mol were compared. With MEK increased to 0.3 mol, the surface area of the cohesive wall of the fibers tended to become smaller, and the depression area measured approximately 10 μm in wide. Moreover, with DEK addition, since the cellulose cohesion was promoted, the fiber cohesion appeared to be stronger, and the pattern of surface depressions was more distinct. This is because of the denser thickness of the cellulose aggregates forming the depressions. In the internal section, in contrast to the surface structure, the cross-sectional image showed a densely agglomerated cellulose structure without gaps. This is due to the fact that the addition of DEK increases the cohesiveness of the cellulose, resulting in a dense gel structure. As a result, in this morphological structure of the hydrogel, particularly in the hydrogel with AC 0.3 and MEK 0.3, an internal depression-like shelf that retains water was observed. The wall thickness of the cellulose fiber that holds water is also thought to contribute to the water retention. Considering that AC and MEK additives influence the hydrogel structure, this would affect the enhancing in water retention in the hydrogel.

For evaluating the effect of additives on the mechanical properties of the resultant hydrogels, both rheological analysis and tensile strength testing were conducted. Viscoelasticity is widely recognized as a valuable method for evaluating the polymer networking of hydrogels [23,24,25]. Figure 6 shows the dynamic (G′) and loss (G″) modulus when subjected to mechanical deformation within the range of 0.1% to 100% strain. Without additives, the value of G′ gradually decreased as the strain% increased from 0.1 to 1%, suggesting a gradual breakdown of the gel under mechanical deformation. The loss modulus value was almost constant during this period. However, the significant decrease in G′ occurred around strain% = 1.3, where G′ = G″ at tan *δ* = 1. This indicates that cellulose is in a liquefied state as G′ < G″ during this strain% interval, resulting from the application of large deformation. In contrast, the addition of ketones did not lead to a decrease in G′ due to mechanical deformation, up to approximately 1% strain. The investigation implies that the observed increase in resistance to deformation is influenced by the homogenization of the gel’s pore structure through ketone addition, as demonstrated in the SEM images (Figure 5). The values of strain% at G′ = G″ are 2.5, 1.7, and 1.3% for the addition of 0.1 mol, 0.2 mol, and 0.3 mol of AC, respectively. The results suggest that the gel state withstands mechanical deformation up to 2.5% strain with the addition of 0.1 mol of AC. With MEK and DEK, the ability to withstand mechanical deformation tended to increase further, e.g., the addition of 0.2 mol resulted in a liquefied state at 6% and 4.5% strain, respectively. In the case of MEK 0.3, the liquefaction transition occurred at 6.5% strain%. Interestingly, in this case, the water content was about 2050%, approximately 1.1 times higher than that of the NA hydrogel. This enhanced flexibility of the gel may have been caused not only by the reinforcement of the shelf-like structure of the water lump surrounding cellulosic fibers but the increased fiber aggregation.

The tensile mechanical properties of hydrogels are shown in Figure 7. The tensile strength was enhanced from 135 kPa to a maximum of 190 kPa, as well as the elongation at break increasing from 102% to 120%, when adding ketones to the solution. The hydrogels prepared by ketone additives exhibited a tendency of increasing tensile strength, especially in MEK 0.3 and 0.2. As a result, the MEK hydrogel demonstrated increased water retention capacity while being mechanically reinforced. This could be attributed to the structural effect observed in SEM images, where the cellulose bank surrounding water clusters might enhance the hydrogel’s strength. These results suggest that the hydrogel presented in this study may extend the future application of physical cellulose hydrogel without chemical modifications in cellulose fiber.

### 2.3. Consideration of the Water-Retaining State of Cellulose Hydrogels

For further understanding, the water retaining state of cellulose hydrogels was investigated using FT-IR with heating, and thermal analysis. Figure 8 shows the FT-IR absorption spectra of cellulose hydrogels in a wet state at various temperatures. The absorption spectra at 25 °C is primarily influenced by the high water content, leading to a broad absorption band of the OH group due to hydrogen bonding [32]. However, as the temperature was gradually increased from 25 °C, the water evaporated, leading the absorption band corresponding to the OH stretching vibration of free water to disappear. The structural cellulose OH band at 3370 cm^−1^ then persisted [33]. As the temperature increased, the NA hydrogels exhibited absorption bands of the cellulose characteristic peaks between 110 and 120 °C. The major change observed from 25 °C to 110 °C involved the absorption band of water hydrogen bonded to the OH group of cellulose. This spectral change in the absorption region occurs similarly in the hydrogel fabricated with additives, mainly in the range of 110 °C to 120 °C. However, the maximum absorption wavenumber of the OH stretching band was found on the longer hydrogel than in cotton at 3370 cm^−1^. In the case of NA, AC, MEK, and DEK, respectively, the values were 3358, 3408, 3398, and 3401 cm^−1^. This result suggested that after dehydration, the OH groups in the cellulose structure form hydrogen bonding networks for the intermolecular interactions of cellulose segments [33,34].

It was reported that in the absorbed water, the hydrophilic polymer develops two types of hydrogen bonds [35,36]: one corresponds to the water molecules directly attached to the active site of the polymer to form the first hydration layer and another corresponds to the water molecules in the second hydration layer. Thus, in the 3000–3700 cm^−1^ region, the retained water via hydrogen bonded in cellulose hydrogels was mainly assigned to the OH group of water. But these waters evaporated due to heating between 50 and 80 °C, leaving behind dehydrated cellulose after reaching over 100 °C. This spectral change in the absorption region occurs similarly with the addition of additives, mainly in the range of 110 °C to 120 °C.

On the other hand, there is the water absorption of crystalline water in the region of 1600 cm^−1^ in the infrared absorption, attributing vibrations of the H-O-H angle [36,37]. In the case of raw cotton fibers, the absorption appears at 1635 cm^−1^ even when IR absorption is measured in the dry state. Within the hydrogels, the water peak at 1640 cm^−1^ gradually diminishes with increasing temperature, becoming nearly imperceptible at 120 °C. On the other hand, MEK 0.2 exhibits a persistent peak attributed to crystalline water even after heating to 110–120 °C, which disappeared significantly between 120 and 130 °C. Therefore, this suggests that the water state in this case is a hydrogel structure with a high percentage of crystalline water as well.

Further analysis of the temperature dependence of the hydrogels was performed using thermal analysis, and the results of thermogravimetric analysis (TGA) and calorimetric differential thermal analysis (DTA) are shown in Figure 9. In the TGA, which shows the weight change at each temperature (top), a significant weight decrease in any of the hydrogels occurs almost entirely between 50 and 80 °C. The change below 100 °C corresponds to the temperature-dependent release of water from the hydrogel due to hydrogen bonding, as indicated by the temperature-dependent relationship of the infrared spectra. On the other hand, the release of crystalline water above 100 °C is almost absent in TGA, indicating that this crystalline water is scarce in quantity. In the calorimetric DTA, an exothermic peak due to dehydration was seen around 75 °C in each case. However, the hydrogels prepared with additives showed an exothermic peak at a temperature a few degrees lower (Table 2).

In contrast, the exothermic peak area for water evaporation is larger in hydrogels prepared with the additive than NA. The value of heat of vaporization (ΔH) corresponds to the amount of water retained in the hydrogels, as seen in Table 2. In other words, the amount of free water retained in the network formed by the cellulose fibers of the hydrogel would depend on the porosity of the hydrogel.

The porosity of these hydrogels was analyzed by nitrogen adsorption experiments on freeze-dried samples. Even though a dried hydrogel is different from a wet hydrogel, we investigated nitrogen absorption experiments in order to analyze the porosity of freeze-dried gels. Figure 10 shows the relationship between the amount of adsorption and the pore diameter at different nitrogen pressures. The addition of the ketones resulted in adsorption isotherms with a sharp rise in gas adsorption in the very low-pressure region, indicating that the additive supported to form micropore space in the hydrogels. According to their adsorption behavior, nitrogen adsorption isotherms are classified by the International Union of Pure and Applied Chemistry (IUPAC) and are designated as Type I [38]. This type of isotherm is characterized by microporosity, signifying the presence of micropores (<2 nm). This suggests that the freeze-dried hydrogels consisted of both micropore and macropore structures.

The pore distribution analysis clearly indicated the presence of micropores, and the pore volume tended to increase with the addition of any of the additives. However, when examining the water content, the pore distribution in AC 0.3 and MEK 0.3, which have higher water content, was almost the same as that of the other samples. In AC, the volume of micropores was lower in AC 0.3 than in AC 0.1, but the water content was higher in AC 0.3. In DEK 0.2, the degree of micropore volume was greater than in AC 0.3, but the water content was less than in AC 0.3 or MEK 0.3, about 1900%, which is similar to that NA hydrogel. Therefore, it appears that the micropores formed by the cellulose fibers do not play a significant role in the retention of water in the hydrogel. Table 2 shows the BET specific surface areas determined by the Brunauer–Emmett–Teller (BET) method. In the BET surface area, the addition of ketones resulted in an increased surface area, reaching approximately 20–26 m^2^/g with a pore size of 5.14–5.53 nm, compared with 17.8 m^2^/g with pore size of 5.39 nm for NA hydrogel. Correspondingly, the pore volume also showed an increasing trend in the cases of hydrogels prepared with additives. These results are also attributed to the fact that the addition of the ketones enhancing the cohesion of cellulose fibers and the porosity of the resulting hydrogel, enabling it to retain large amounts of water within its structure.

## 3. Conclusions

Hydrogels were prepared from DMAc/LiCl solutions of cotton cellulose with AC, MEK, and DEK as additives, and their properties were investigated in this study. The presence of the additives was found to promote the cohesive nature of the cellulose fibers in the solution, which improves the water retention properties of the resultant hydrogels, especially in the cases of AC and MEK, and also results in a gel structure that withstands the impossibility of mechanical stresses in the hydrogel. The aggregated cellulose network of water-retaining fibers forms an enveloping structure in a tufted pattern, as shown in Figure 11, and the gel structure is supported by aggregated thick cellulose fibers with a mesh structure which can retain high water content, while providing responsiveness that can maintain the gel structure against external mechanical properties. We believe that the cellulose hydrogel fabricated with ketone additives might have an application in the medical field. Further application studies are needed in the future to confirm the effectiveness.

## 4. Materials and Methods

### 4.1. Materials and Chemical Reagents

Defatted cotton was purchased from Kawamoto Corporation (Osaka, Japan) and used as a cellulose source. Ethanol, *N*, *N*-dimethylacetamide (DMAc), lithium chloride (LiCl), acetone (AC), methyl ethyl ketone (MEK), and diethyl ketone (DEK) were purchased from Nacalai Tesque, Inc. (Tokyo, Japan). DMAc was dried with molecular sheaves (4A 1/8) at room temperature for over three days, and LiCl was dried at 80 °C in a vacuum before use.

### 4.2. Preparation of Cellulose Solutions and Hydrogels

Prior to the preparation of the DMAc/LiCl solution, the cotton cellulose was subjected to three different solvent exchange processes of water, ethanol, and DMAc for one day each. Then, 1 wt% cotton was dissolved in 6 wt% LiCl concentration of DMAc solution to obtain a 1 wt% cellulose solution. Then, 0.1–0.3 mol (per 1 g cellulose) of AC, MEK, or DEK was added to the prepared cellulose solution, respectively. The solutions were stirred at room temperature until the solutions were completely mixed and the coagulated cellulose fibers were completely redissolved. The solution was poured into a glass Petri dish (5 cm diameter × 1 cm depth) with 10 g, then put in plastic container (11 cm × 11 cm × 5 cm height) with 10 mL ethanol and kept under the ethanol vapor atmosphere for 24 h at room temperature, allowing phase inversion gelation. After the gel films were obtained, the films were washed using excess distilled water and immersed in water for 24 h to remove remaining DMAc and LiCl.

### 4.3. Characterization of Cellulose Solutions and Hydrogel Films

Solution viscosity and dynamic viscoelasticity of hydrogels were measured with a rheometer (Physica MCR301, Anton Paar, Graz, Austria). Temperature control was ensured by means of a Peltier element (PTD-200) for keeping temperature at 25 °C. The shear viscosity of the cellulose in DMAc/LiCl solution was measured with a cone plate’s geometry; diameter: 25 mm; angle: 2°. A total of 0.5 mL of fluid sample was used for each test measurement. When varied with the shear rate in the range from 0.1 to 1000 s^−1^, the shear viscosity was recorded. The gelation time of cellulose in an ethanol vapor atmosphere was determined by oscillatory shear experiments using a parallel plate’s geometry with a diameter of 25 mm. When storage (G′) and loss (G″) elastic moduli were measured for the cellulose-DMAc/LiCl solution, 10 g solution was filled in glass Petri dish with 35 mm diameter and the parallel plate rotor was set in the solution. As the dish was placed in plastic container with 5 mL of ethanol and covered, the strain of 1 Pa at a frequency of 1 Hz was applied at the gap between the plates of 1 mm for the moduli measurements. The viscoelastic properties of hydrogel were investigated using the rheometer with a parallel plate’s geometry. When deformation was applied by different strain% to the hydrogel, both G′ and G″ elastic moduli were measured in the strain sweep from 0.01 to 100% at a frequency of 1 Hz.

Water content of cellulose hydrogel was calculated on a dry weight basis as follows: The weight of swollen samples was measured after wiping off the remaining surface water. Afterward, the samples were dried with vacuum for 24 h to acquire the dried weight and then the values of water contents was calculated based on the following formula: water contents [%] = ((W_s_ − W_d_)/W_d_)× 100), where W_s_ was the weight of the swollen hydrogel and W_d_ was the weight of the dried hydrogel. The reported water content was an average of triplicate. The morphological analysis was studied by using TM3030Plus tabletop microscope (HITACHI High Tech, Tokyo, Japan) for wet and freeze-dried films. The tensile strength of hydrogel films was carried out using a Tensile and Compression Testing Machine (LTS-500-S20, Minebea, Tokyo, Japan). Films were cut into 30 × 10 mm test pieces with a gauge length of 10 mm. Testing was conducted at room temperature and 5 mm/min strain rate.

The temperature dependence of FT-IR spectra was studied by using IRPrestige-21 (Shimadzu Co., Ltd., Tokyo, Japan) in the range from 4000 to 1000 cm^−1^ at 2 cm^−1^ resolutions with 32 scans for each piece. Samples were sandwiched by CaF_2_ plates. The temperature was controlled using program temperature controller (TXN-700B, ASKUL Corp., Tokyo, Japan) in the range of 30–180 °C. Thermal behavior of hydrogels was studied using a TG-DTA (TG-DTA81022, Rigaku, Tokyo, Japan) at a heating rate of 5 °C/min from 30 up to 600 °C under the air flowing. Pore size distribution of the cellulose hydrogel films was determined from nitrogen adsorption and desorption isotherm at 77 K using TriStar II 3020 (Micromeritics, Norcross, Georgia, USA). The mesopore size distribution was calculated by the Barrett–Joyner–Halenda (BJH) model applied to the adsorption branch of the isotherm. Before the measurement, swollen hydrogel films were freeze-dried using freeze dryer (Lyovapor^TM^, Nihon BUCHI K.K., Tokyo, Japan).

## Figures and Tables

**Figure 1 gels-10-00034-f001:**
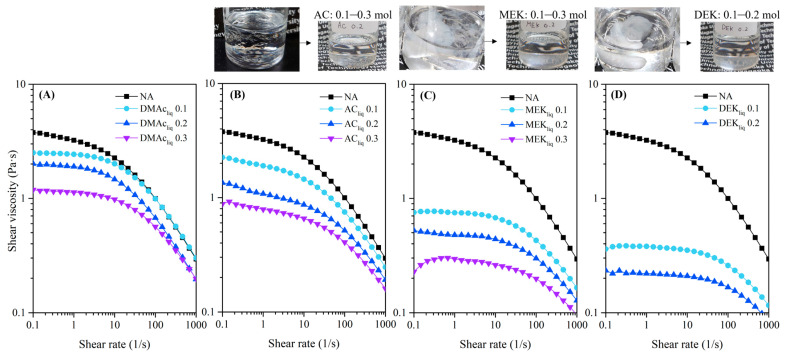
Shear viscosity at shear rate for cotton cellulose solution in DMAC/LiCl in the presence of DMAc (**A**), acetone (**B**), methyl ethyl ketone (**C**), and diethyl ketone (**D**).

**Figure 2 gels-10-00034-f002:**
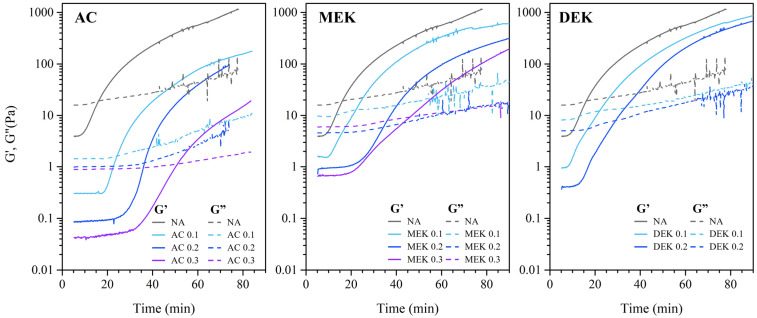
Time variation in storage modulus (G′) and loss modulus (G″) for cotton cellulose DMAc/LiCl solution exposed in ethanol vapor atmosphere at 25 °C.

**Figure 3 gels-10-00034-f003:**
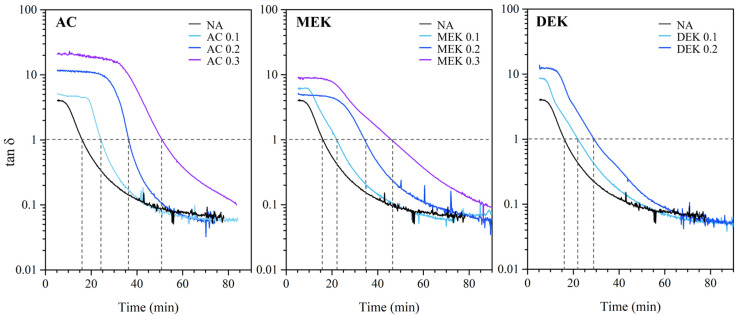
Time variation of tan *δ* for each DMAc/LiCl solution.

**Figure 4 gels-10-00034-f004:**
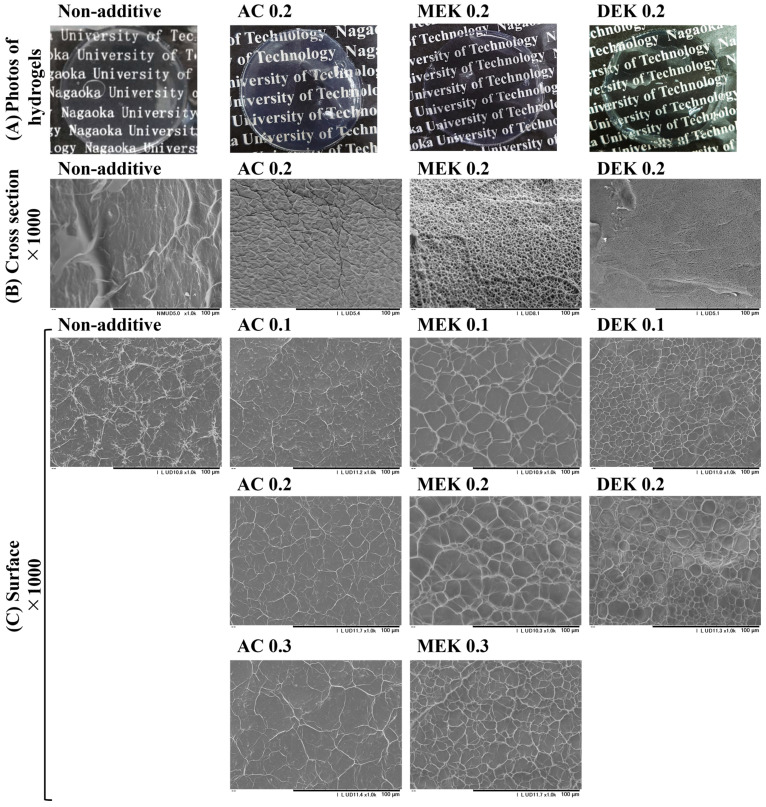
Pictures of hydrogel appearance (**A**) and SEM images of cross section (**B**) and surface (**C**) of resultant hydrogels.

**Figure 5 gels-10-00034-f005:**
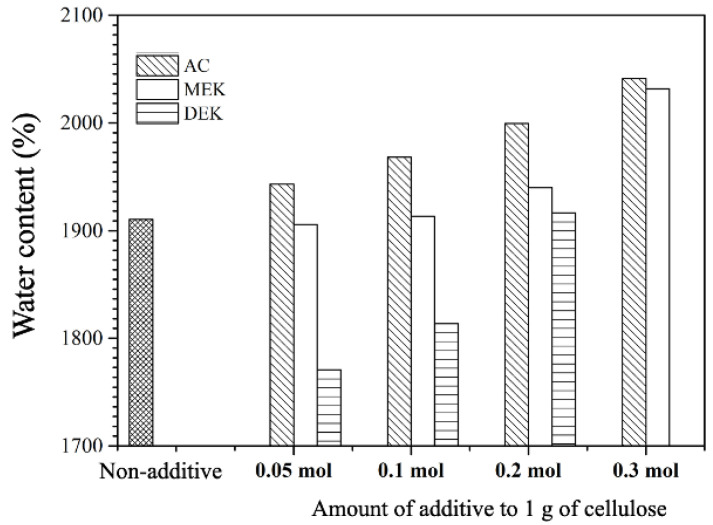
Water content of hydrogel made from cellulose solution of additives added to 1 g cellulose.

**Figure 6 gels-10-00034-f006:**
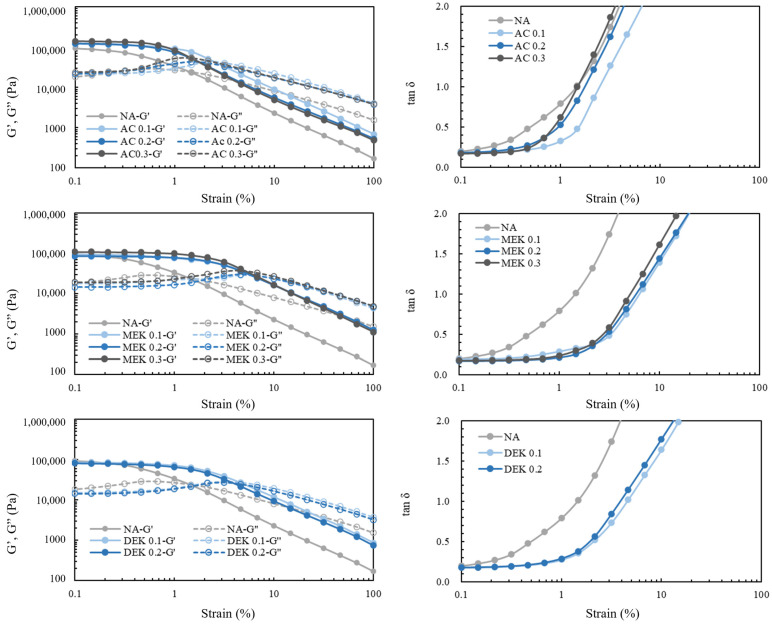
G′ and G′′ moduli of cellulose hydrogels and tan *δ* at different strain%.

**Figure 7 gels-10-00034-f007:**
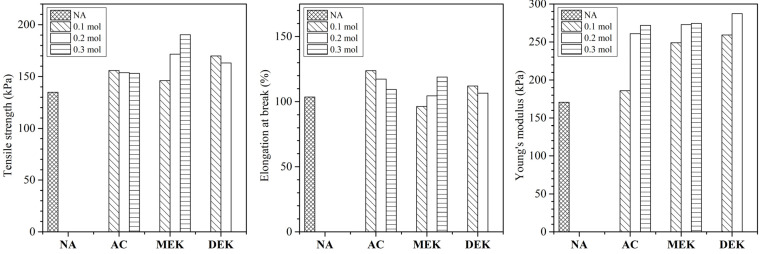
Tensile strength, elongation at break and young’s modulus of cellulose hydrogels.

**Figure 8 gels-10-00034-f008:**
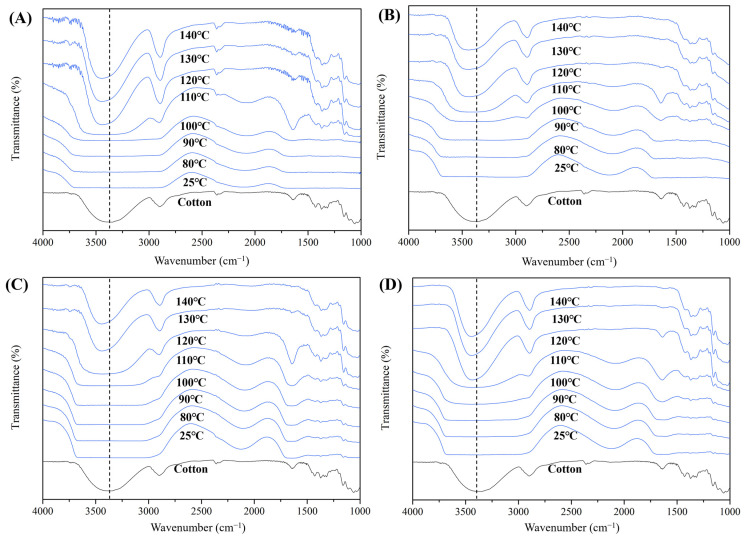
FT-IR spectra of cotton cellulose hydrogels with water as varying temperature from 25 °C to 140 °C (**A**) Non-additive, (**B**) AC0.2, (**C**) MEK0.2, (**D**) DEK0.2.

**Figure 9 gels-10-00034-f009:**
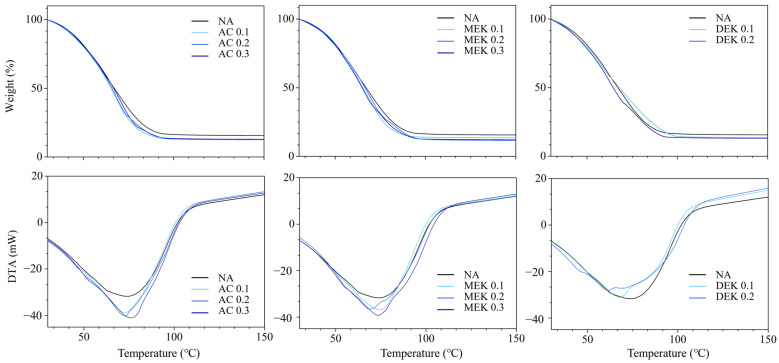
Thermogravimetric analysis (**upper**) and calorimetric differential thermal analysis (**lower**) for cotton cellulose hydrogel with water.

**Figure 10 gels-10-00034-f010:**
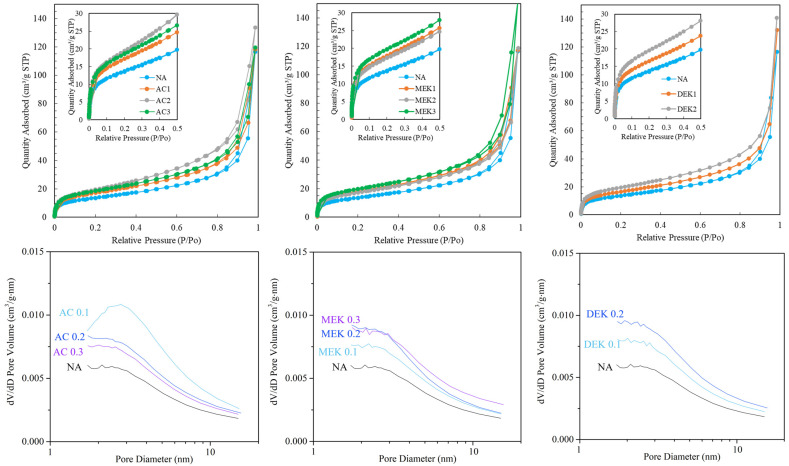
N_2_ adsorption of dehydrated cellulose hydrogels at each relative pressure (**A**) and pore distribution in the relationship between pore volume and the diameter (**B**).

**Figure 11 gels-10-00034-f011:**
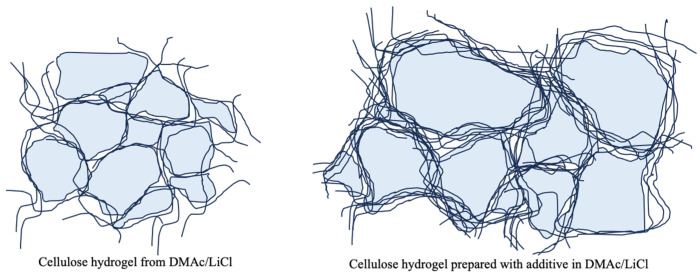
Illustration of encapsulated water in the water retention state of hydrogels prepared with and without additives in cotton cellulose fiber network consisting of agglomerated fibers.

**Table 1 gels-10-00034-t001:** Properties of cotton cellulose–DMAc/LiCl solutions.

	Concentration (mol/1 g Cellulose)	Shear Viscosity ^a^ (Pa·s)	Gelation Time (Min)
Non-additive	0	3.49	16.1
AC 0.1	0.1	1.99	24.3
AC 0.2	0.2	1.10	36.4
AC 0.3	0.3	0.96	51.0
MEK 0.1	0.1	1.88	22.1
MEK 0.2	0.2	0.69	34.2
MEK 0.3	0.3	0.66	46.0
DEK 0.1	0.1	1.57	17.1
DEK 0.2	0.2	0.60	29.1
DEK 0.3	0.3	n/a	n/a

^a^: Shear viscosity at a shear rate of 0.1 s^−1^.

**Table 2 gels-10-00034-t002:** Parameter values of thermal analysis and N_2_ adsorption on the dehydrated hydrogels.

Sample	Water Evap.		BET Surface Area	Pore Size	Pore Volume
	Temp. (°C)	Δ*H* (J/g)	(m^2^ g^−1^)	(nm)	(cm^3^ g^−1^)
Non additive	75.1	−1438	17.8	5.3855	0.051
AC 0.1	72.7	−1490	20.1	5.2061	0.062
AC 0.2	73.5	−1493	24.9	5.2091	0.087
AC 0.3	76.4	−1496	25.5	5.2548	0.079
MEK 0.1	74.5	−1463	21.6	5.1385	0.067
MEK 0.2	68.3	−1531	20.1	5.3023	0.062
MEK 0.3	73.6	−1567	26.1	5.5351	0.078
DEK 0.1	61.5	−1304	21.5	5.2681	0.064
DEK 0.2	68.4	−1411	25.2	5.2924	0.076

## Data Availability

The data that supports the findings of the current study are listed within the article.

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
