# Peer review of "Phase Inversion Gelation Process and Additive Effects on Hydrogel Film Properties of Cotton Cellulose"

_gels, 2023, doi:10.3390/gels10010034_

Round 1

Reviewer 1 Report

Comments and Suggestions for Authors

The manuscript describes fabrication and properties of hydrogel film of cotton cellulose using well-known phase inversion gelation process and acetone, methyl ethyl ketone and diethyl ketone as additives. The fundamental background of the manuscript is not very novel, effect of different additives on cohesion of cellulose fibers and hydrogel structure have been earlier addressed in the literature. The authors have demonstrated that mechanical properties of hydrogels can be improved by methyl ethyl ketone and diethyl ketone addition. However,  taking into account potential applications mentioned in the introduction,   tensile stress-strain curves of hydrogels with different  doses of additives would be of significant benefit.  

Description of the results and conclusions are difficult to follow due to the rather poor English. Sometimes, one cannot even understand what exactly the authors wanted to say. They also  use excessively long descriptions for the plots, which are self-explanatory. Especially, this does not make any sense for porosity measurements. The terminology needs revision, e.g. vapor/steam are used as synonyms that leads to the confusion about way of the samples preparation.

Comments on the Quality of English Language

English is rather poor. Sometimes one cannot understand the ideas of the authors.

Author Response

The point by point answers for Reviewer 1 is attached by pdf file.

Reviewer 2 Report

Comments and Suggestions for Authors

Esteemed authors of the manuscript provided an interesting paper on widespread cellulose hydrogels (HGs) prepared from solutions of cotton cellulose in DMAc/LiCl. By now dozens, if not hundreds, of articles have been published with a variation of this theme. Nevertheless, it is particularly interesting what new concept the authors have applied in studying the process and properties of the obtained HGs.

The abstract erroneously states the solvent system. The text discusses a different.

·        Even after reading just Introduction, a number of questions arise. Some of them are cardinal. The first one relates to the authors' choice to apply cotton cellulose for their experiments. The authors' argument that global cotton production is increasing is not convincing for a simple reason: cotton is substantially more expensive than other types of plant-based pulp.

·        In addition, cotton cellulose is known to dissolve in solvent system DMAc/LiCl to a lesser extent and dissolution needs more time than other celluloses. This is exactly what the authors themselves prove in the experimental part, as it is described that the mixture cellulose:solvent (about 1% concentration) needs stirring „for 1 month to obtain a clear colorless cotton cellulose solution“??!.

·        Moreover, the phrase that „the cellulose solution is brought into contact with water vapor or alcohol vapor, which reduces the solubility of the cellulose and results in gelation as the water vapor concentration in the DMAc/LiCl solution increases, a wet phase conversion method, resulting in the phase transformation from a liquid to a gel-like solid.“ is not quite clear. It is also not an argument in favour of the wet phase conversion method, since coversion of a solution into a gel-like solid also occurs under normal conditions in an air atmosphere.

·        The use of additives from a range of ketones is also not argued and reduces the "green" component of this study.

There are questions to Results and discussion.

·        When the effect of adding the additives is discussed, the authors state the well-known fact that diluting a solution of cellulose in DMAc/LiCl with any additives that do not dissolve cellulose and are cellulose precipitators ensures that cellulose aggregates and precipitates out of the solution.

·        There is also no clarification to explain the difference in the effect of the additives that belong to one class of compounds on viscosity and its variation of solutions.

·        The same questions arise to the paragraph of storage modulus and loss modulus as well as to the paragraph on water content in the hydrogels are described.

·        Regarding the SEM study of the obtained hydrogels, the photo non-additive hydrogel with obvious aggregation can be explained by the properties of the unmodified cellulose used to obtain the hydrogel. Obviously, the formation of hydrogels from the solutions without dilution and diluted and the properties of hydrogels will be different, as the solubility in solutions and the conditions of the hydrogel formation are different.

There is no need to look in detail at the other sections of the manuscript, as they are all described in the same way.

The manuscript is professionally written and covers a wide range of complex issues. This is the reason why not all of them have been dealt with in depth, so it needs to be revised for a more in-depth consideration, possibly with a reduction in the issues addressed.

Conclusion. The manuscript needs to be revised in the light of the comments.

Author Response

The point by point answers for Reviewer 2 is attached by pdf file.

Reviewer 3 Report

Comments and Suggestions for Authors

The manuscript by Ibaraki & Kobayashi is devoted to the effect of the addition of a number of ketones on the structural and mechanical properties of cellulose hydrogels. The authors examined in detail the change in the viscosity and viscoelastic properties of hydrogels, studied the morphology of the hydrogel surface and cross-section, as well as the state of water inside the gels. In addition, the authors studied the porosity of the structure, surface area and showed the presence of micro- and macropores in the samples. It has been shown that micropores do not play a significant role in water sorption.

Research makes a pleasant impression; it is logically executed and each method used by the authors answers a question arising from the previous study.

However, the authors did not indicate the pore size obtained during the BET study, providing only data on the presence of micro- and macropores. What is the pore sizes? As far as I know, pores larger than 500 nm are difficult to analyze using the BET method. Was there any additional sample preparation performed besides lyophilization?

The authors should also replace Figures 1 and 2 with their own higher resolution ones. And it is not clear what the authors wanted to say with the top row of images in Figure 5.

Author Response

The point by point answers for Reviewer 3 is attached by pdf file.

Reviewer 4 Report

Comments and Suggestions for Authors

Comments and Suggestions for Authors

 Dear authors, I have read your article entitled,

Phase inversion gelation process and additive effects on hydrogel film properties of cotton cellulose.

My observations are the follows:

·       Develop the general objective of the study, highlight the novelty and originality of the study.

·       Hydrogels were prepared from DMAC/LiCl solutions of cotton cellulose with AC, 322 MEK, and DEK as additives, and their properties were investigated in this study.  I don't see any practical application of these hydrogels. They are used for?

·       Add preliminary conclusions to each experiment presented.

·       In the Conclusion section, it is necessary to indicate the direction of further research, as well as the possibility of practical use of the data obtained.

Author Response

The point by point answers for Reviewer 4 is attached by pdf file.

Round 2

Reviewer 1 Report

Comments and Suggestions for Authors

The manuscript was substantially improved and can be accepted. 

Comments on the Quality of English Language

English requires extensive editing. 

Author Response

The answer for Reviewer 1 is attached by PDF file.

Reviewer 4 Report

Comments and Suggestions for Authors

Dear authors, I have read your answers. Please add what you replied to me in the article.

Some of the answers are only given in the review report to reviwer. 

The role of these questions was to help you to improve the quality of the article.

Author Response

The answer for Reviewer 4 is attached by PDF file.
